# Hierarchical Clustering and Multivariate Forecasting for Health Econometrics

Atika Rahman Paddo
apaddo@iu.edu
Indiana University Purdue University
Indianapolis
Indianapolis, Indiana, USA

Sadia Afreen
fnsadia@iu.edu
Indiana University Purdue University
Indianapolis
Indianapolis, Indiana, USA

Saptarshi Purkayastha
saptpurk@iupui.edu
Indiana University Purdue University
Indianapolis
Indianapolis, Indiana, USA

## ABSTRACT

Data science approaches in Health Econometrics and Public Health research are limited, with a lack of exploration of state-of-the-art computational methods. Recent studies have shown that neural networks and machine learning methods outperform traditional statistical methods in forecasting and time-series analysis. In this study, we demonstrate the use of unsupervised and supervised machine learning approaches to create "what-if" scenarios for forecasting the long-term impact of changes in socio-economic indicators on health indicators. These indicators include basic sanitation services, immunization, population ages, life expectancy, and domestic health expenditure. To begin, we utilized Hierarchical Cluster Analysis to group 131 countries into 9 clusters based on various indicators from the World Bank Health Statistics and Nutrition dataset. This step allowed us to create clusters of countries. In order to showcase the feasibility of our approach, we performed a time series analysis using multivariate prophet on the most significant features from a cluster consisting of Bahrain, Kuwait, Oman, Qatar, and Saudi Arabia. The study developed robust models ($R^2$ = 0.93+) capable of forecasting 11 health indicators up to 10 years into the future. By employing these "what-if" scenarios and forecasting models, policymakers and healthcare practitioners can make informed decisions and effectively implement targeted interventions to address health-related challenges.

## CCS CONCEPTS

• **Computing methodologies** → **Modeling methodologies**; • **Applied computing** → **Health informatics**; • **Information systems** → *Clustering*; *Information systems applications*.

## KEYWORDS

Clustering, forecasting, health econometrics, data science

**ACM Reference Format:**
Atika Rahman Paddo, Sadia Afreen, and Saptarshi Purkayastha. 2023. Hierarchical Clustering and Multivariate Forecasting for Health Econometrics. In *Proceedings of epiDAMIK @ SIGKDD Workshop.* ACM, New York, NY, USA, 8 pages. https://doi.org/XXXXXXX.XXXXXXX

## 1 INTRODUCTION

Health econometrics is a multidisciplinary field that combines economics and statistics to study various aspects of healthcare systems, policies, and outcomes. Traditionally, econometric methods have been employed to analyze healthcare data, including regression models, panel data analysis, and instrumental variable techniques [20, 7]. However, there is a growing recognition of the potential benefits of incorporating these advanced techniques into health econometrics research.

In today's interconnected society, understanding the factors that affect health outcomes is crucial for effective policymaking and healthcare treatments. With the availability of extensive health data, advanced analysis methods can provide valuable insights to support evidence-based decision-making. The World Bank's Health Statistics collection offers a wealth of data on various health indices across nations [26]. In this study, we aim to develop a better understanding of the predefined Gulf Cooperation Council (GCC) countries, which share similar economies and development goals [15]. By utilizing a clustering algorithm, we have identified similarities in their health statistics [34]. However, this study does not include one of the GCC countries, the United Arab Emirates (UAE).

Katoue et al. argued that the health issues faced in the Middle East and North Africa regions must be highlighted, as these countries still face challenges in providing equitable and high-quality healthcare services. Limited literature supports evidence of improvements in these areas [13]. To address the health challenges in the GCC countries, including Bahrain, Kuwait, Oman, Qatar, Saudi Arabia, and the UAE, innovative strategies are necessary to improve the overall health status of the Middle Eastern countries [15, 19]. A United Nations report highlights disparities and commonalities in health factors among different regions in the Arab world [31]. While the report suggests that the GCC countries have made progress in maintaining sanitation and safe drinking water, it is unclear whether all countries in the region will continue with the same policies in the future [31].

This study aims to identify any disparities between countries regarding uniform healthcare provision. The 2015 World Bank report emphasizes the impact of health outcomes on health policies and expenditure in the GCC countries [28]. Changes in health outcomes, such as non-communicable diseases and life expectancy, coupled with inflation, may create disparities in health expenditure among these countries [2].

It remains uncertain which countries can improve overall healthcare and which may lag behind in developing uniform health policies [8]. Additionally, our research study focuses on population well-being, particularly in different age groups, and factors such

as expenditure, immunization, and survival rates. Understanding the association between age and other health factors is crucial for targeting "age-specific" policies in healthcare management and disease prevention [9]. This is significant in terms of healthcare management and disease prevention.

This research paper combines cluster analysis, feature importance analysis, and multivariate time series modeling to uncover the underlying factors influencing health outcomes within a selected cluster comprising five GCC countries: Bahrain, Kuwait, Oman, Qatar, and Saudi Arabia. The findings contribute to a deeper understanding of the complex dynamics of health indicators and provide actionable insights for policymakers and healthcare professionals.

## 2 RELATED WORKS

Balçik et al. [5] conducted a study on clustering algorithms that is similar to ours. They focused on the hierarchical clustering of European Union countries based on preselected features to analyze healthcare development. Their clustering results were evaluated using statistical differences between indicator values. Similarly, Raheem et al. [29] approached their objective using the silhouette score, providing a clearer context for distinguishing clusters. While both approaches seemed reasonable, we opted to use the silhouette score in our study to understand the distinctiveness of our clusters, which yielded high accuracy in identifying cluster formation.

Several studies have been conducted on a national level using clustering approaches to determine differences in health indicators and gain insights into various countries. Proksch et al. [27] analyzed the clustering of 30 OECD countries to identify the varying aspects of health that differentiate these clusters. Muldoon et al. [23] and Lefèvre et al. [17] explored similarities among countries and their contributions to health factors. The former focused on mortality significance, while the latter employed a multivariate clustering approach to identify patterns in population and healthcare systems. In contrast to these studies, our research includes a forecasting approach, which provides predictive conclusions for policymakers, analysts, and health practitioners.

Levantesi et al. [18] also utilized a multivariate forecasting approach to develop a predictive understanding of healthcare, albeit not aligned with the Prophet model. Khan & Noor [14] explored the application of the Prophet time series approach to visualize future health outcomes, but their study employed a univariate Prophet approach. In our study, we employed a multivariate Prophet approach, which offered a unique perspective by determining the relationship between changes in one indicator and another more accurately. Ahmed et al. [1] and Ampofo & Boateng [4] also adopted interesting approaches using multivariate Prophet, focusing specifically on cardiovascular and diabetes health sectors, respectively.

Therefore, our research aims to establish a comprehensive association among predicted population well-being, which can be utilized to advance our understanding of healthcare outcomes.

## 3 METHODOLOGY

The methodology utilized in this research paper followed a sequential process to analyze health data. Firstly, the data underwent preprocessing. Next, a dendrogram was constructed using the Ward

method to identify clusters. A threshold was applied using the 'fcluster' function to determine the number of clusters. Afterward, the important features for each cluster were identified using a threshold of 0.615. We employed the multivariate Prophet method for time series forecasting and predicting future trends. Finally, statistical tests were conducted on the features to identify significant differences in the upcoming years.

### 3.1 Data Collection

We obtained the Health Statistics and Nutrition dataset from The World Bank, which offers comprehensive health indicators for various countries from 1960 to 2021.

### 3.2 Data Preprocessing

*3.2.1 Data Cleaning.* Initially, the original dataset contained information for 266 countries/regions and 255 indicators. To focus on a specific midway time shot, we selected data from 2000. We excluded regional aggregations from the dataset (EU, AFRO, etc.) and countries with significant missing values for most indicators (e.g., United Arab Emirates, Aruba, Afghanistan, Poland, Barbados, Guinea). Additionally, we removed indicators with extensive null values across countries. Any remaining null values for a country were imputed using the median of that column. After cleaning, the dataset comprised 134 countries and 128 variables.

*3.2.2 Data Scaling using Min-Max Scaler.* To ensure consistency and prevent any single feature from dominating the analysis, we scaled the data using the Min-Max Scaler [6]. This scaling technique transformed the data to a predefined range of 0 to 1 by subtracting the minimum value and dividing by the range. This process normalized the data within the [0, 1] range.

### 3.3 Clustering

*3.3.1 Linkage Matrix.* Next, we computed the linkage matrix using the linkage function from the *scipy.cluster.hierarchy* module. The linkage matrix represents the hierarchical clustering structure of the data based on pairwise distance calculations.

*3.3.2 Creating a Dendrogram using Ward's Method.* We employed Ward's method to construct a dendrogram, which visually displays the hierarchical relationships among the data points [24]. Ward's method minimizes the total within-cluster variance at each step of dendrogram creation. The resulting dendrogram exhibited hierarchical clustering patterns from a distance scale of 0 to 27, aiding in understanding the grouping patterns within the data (see Fig. 1).

*3.3.3 Determining the Number of Clusters using fcluster.* The number of clusters was determined by assigning data points to clusters based on a given threshold using the *fcluster* function. A threshold value of 5 was chosen to define the clusters within the dataset. The *fcluster* function, with the specified threshold, provided the cluster assignments for each data point. The above threshold resulted in 9 clusters.

*3.3.4 Evaluation Metrics for Each Cluster:* To assess the quality of the clustering results and evaluate the fit of each data point to its assigned cluster, we calculated the Silhouette score for each cluster. The Silhouette score measures both the cohesion within

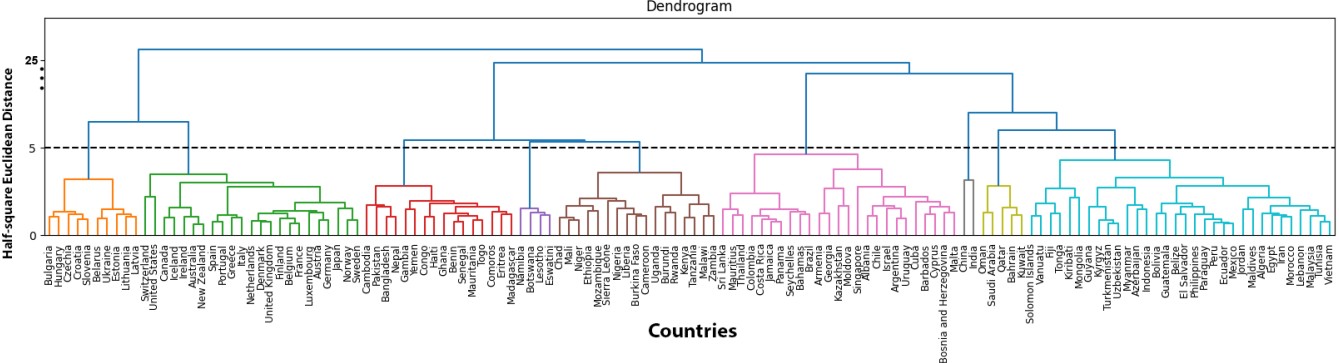

**Figure 1: Linkage matrix of nine clusters for the countries in a dendrogram**

each cluster and the separation between clusters [32, 25]. The score was calculated using equation 1.

$$Silhoutte = \frac{\sum \frac{b_i - a_i}{max(a_i, b_i)}}{n} \qquad (1)$$

where, $a_i$ is the average distance between each sample for $i = 1, 2, 3, ...n$ and all other points in its cluster. For each other cluster in the dataset, the average distance between the sample and all points in that cluster is noted and the minimum of these distances is $b$. $n$ is the total number of samples. To calculate per cluster Silhouette score, $a$ represents the average distance between the data point and other data points within the same cluster and $b$ represents the average distance between the data point and the data points in the nearest neighboring cluster.

The Silhouette score ranges from -1 to 1, with a higher score indicating better clustering results. A score close to 1 signifies well-separated clusters, while a score close to -1 suggests overlapping or incorrectly assigned clusters. The average silhouette score of all data points within the cluster was calculated to obtain the silhouette score for each cluster. Based on the silhouette score and more attainable count of the cluster, cluster-8 was chosen for further analysis of time series forecasting.

*3.3.5 Using hierarchical clustering over other clustering methods:* We chose hierarchical clustering using Ward's method for our analysis of the health statistics and nutrition dataset. Hierarchical clustering allows us to explore the data in a hierarchical structure, capturing both global and local patterns of similarity. It is well-suited for datasets with arbitrary cluster shapes and sizes, making it suitable for analyzing health indicators across countries.

## 3.4 Feature Selection

Following the clustering of the countries, our focus shifted to pinpointing the most crucial characteristics. We accomplished this by implementing the sklearn library to perform feature selection. We evaluated 26 key features within the selected cluster, which ranked within the top percentile (Table 1).

*3.4.1 Feature Importance Analysis for Each Cluster.* Centroids, or representative data points for each cluster, were determined by averaging the scaled data. The significance of each feature was

ascertained by arranging the feature values in descending order. A threshold of 0.815 yielded fewer features and did not provide a comprehensive outlook for health predictions. As a result, we opted for a threshold of 0.615, which allowed us to conduct a time series forecast with a broader feature set.

## 3.5 Statistical Tests

Our reference timeframe was set to the year 2000 for initiating the time series forecast, and we examined the data for each indicator within the clustered countries. The Kruskal Wallis non-parametric test served as an effective method for determining value significance [36]. We utilized this test to discern statistically significant discrepancies among the indicators' values across different countries. After projecting the values for the next decade (2022-2031), we repeated the statistical test on these forecasted values to highlight significant differences between countries.

## 3.6 Time-Series Forecasting

*3.6.1 Data Processing for Time-Series Analysis.* Several factors were considered when preparing this data for modeling.

*Selection of Time Frame:* To forecast future health statistics for the clustered countries, we opted for the most recent data to train the multivariate Prophet model. Our dataset encompassed health data from 1960 to 2021, but for our purposes, we narrowed the timeframe to 2000 to 2021. This eliminated the need for imputing data from distant years.

*Reduction of Features:* The initial feature importance analysis identified 26 features for the study. However, two features (Cause of death, by non-communicable diseases (% of total) and International migrant stock (% of population)) had a high percentage of missing values across all clustered countries, accounting for up to 81.82% of total data. That is why excluded these indicators and kept 24 features.

*Imputation of Time-series Data:* We identified missing values within our set of 26 features, necessitating imputation for a complete time-series dataset. We used Naïve forecasting to fill in the missing data for the years from 2000 to 2021. If a specific year's data was missing for a particular country's indicator, we filled the gap using

## Table 1: FEATURE IMPORTANCE FOR CLUSTER

| | Indicator Name | Indicator Code | Feature Importance Value |
|---|---|---|---|
| 1 | People using at least basic sanitation services (% of population)[§] | SH.STA.BASS.ZS | 0.9743 |
| 2 | Immunization, measles (% of children ages 12-23 months) | SH.IMM.MEAS | 0.9606 |
| 3 | People using at least basic drinking water services (% of population) | SH.H2O.BASW.ZS | 0.9585 |
| 4 | Immunization, DPT (% of children ages 12-23 months)[§] | SH.IMM.IDPT | 0.9257 |
| 5 | Survival to age 65, male (% of cohort) | SP.DYN.TO65.MA.ZS | 0.8753 |
| 6 | Survival to age 65, female (% of cohort)[†] | SP.DYN.TO65.FE.ZS | 0.8752 |
| 7 | Population ages 25-29, male (% of male population) | SP.POP.2529.MA.5Y | 0.8583 |
| 8 | Population ages 20-24, female (% of female population) | SP.POP.2024.FE.5Y | 0.8437 |
| 9 | Life expectancy at birth, total (years)[†] | SP.DYN.LE00.IN | 0.8227 |
| 10 | Population ages 25-29, female (% of female population)[†] | SP.POP.2529.FE.5Y | 0.8216 |
| 11 | Life expectancy at birth, female (years)[†] | SP.DYN.LE00.FE.IN | 0.7954 |
| 12 | Population ages 30-34, male (% of male population) | SP.POP.3034.MA.5Y | 0.7651 |
| 13 | Cause of death, by non-communicable diseases (% of total)[*] | SH.DTH.NCOM.ZS | 0.7567 |
| 14 | Population ages 20-24, male (% of male population)[§] | SP.POP.2024.MA.5Y | 0.7527 |
| 15 | Population ages 30-34, female (% of female population) | SP.POP.3034.FE.5Y | 0.7318 |
| 16 | Population ages 15-64, male (% of male population) | SP.POP.1564.MA.ZS | 0.722 |
| 17 | Population ages 15-64 (% of total population)[†] | SP.POP.1564.TO.ZS | 0.7077 |
| 18 | Domestic general government health expenditure (% of current health expenditure) | SH.XPD.GHED.CH.ZS | 0.7007 |
| 19 | Population ages 35-39, male (% of male population) | SP.POP.3539.MA.5Y | 0.6914 |
| 20 | Population growth (annual %)[¶] | SP.POP.GROW | 0.689 |
| 21 | International migrant stock (% of population)[*] | SM.POP.TOTL.ZS | 0.6842 |
| 22 | Population ages 05-09, female (% of female population) | SP.POP.0509.FE.5Y | 0.6734 |
| 23 | Population ages 10-14, female (% of female population)[†] | SP.POP.1014.FE.5Y | 0.6686 |
| 24 | Population ages 0-14, female (% of female population)[†] | SP.POP.0014.FE.ZS | 0.6615 |
| 25 | Population, male (% of total population)[†] | SP.POP.TOTL.MA.ZS | 0.6595 |
| 26 | Population ages 15-19, female (% of female population)[†] | SP.POP.1519.FE.5Y | 0.6388 |

[*] Removed because of having 81.82% values as missing from the year 2000 to 2021.

[†] Removed because of having highly correlation with other important feature(s) which were in higher rank according to feature importance.

[§] Removed for poor predictions from the univariate Prophet model and were not used in multivariate model training.

[¶] Removed because of having negative values in some years, thus log transform scaling could not be done, thus removed in the forecasting.

the preceding year's data for that same indicator. This resulted in a complete time-series dataset with 24 features for five countries.

*Logarithmic Scaling on Time-series Data:* Prior to forecasting, we performed a logarithmic transformation for data scaling and reverted to the original values for performance measurement. Although the MinMax Scaling algorithm was used initially, we chose logarithmic scaling for the time series forecast. This decision was based on the lower error rate found with logarithmic scaling when returning to the original data [20].

### 3.6.2 *Prophet Forecasting Model to Predict Indicator Values.* Our approach to predicting yearly indicators' values for the clustered countries and important features involved using multivariate modeling in Prophet. This is what enables "what-if" analysis for forecasting health indicators. If we simulate or forecast individual predictor indicators and guide policy, we can see the effects of those simulations on our final multivariate model. This is crucial to understand how these indicators' forecasts varied per country and whether the Prophet model's results were consistent for all clustered countries.

*Univariate Prophet Model.* The univariate Prophet model focuses on forecasting a single time series taking into account the historical values of the target variable and identifies patterns and trends to make future predictions. The model captures seasonality ($s(t)$), trend ($g(t)$), holiday effects ($h(t)$) (if any) and error ($\epsilon(t)$) using additive regression components.

$$y(t) = g(t) + s(t) + h(t) + \epsilon(t) \qquad (2)$$

In our work, we have used a Univariate Prophet model to forecast the predictor values for the future. However, if existing econometric models of varied types are more suited for a particular indicator, then those can also be used. The univariate model for each predictor built the future dataframe for the years 2022 to 2031 (10 years).

*Multivariate Prophet Model.* The multivariate Prophet model extends the univariate model by incorporating additional exogenous variables or features as regressors that can influence the target variable. These additional exogenous variables ($f_1(t), f_2(t), ..., f_n(t)$) can be other time series data or external factors such as economic indicators. In this work, we have incorporated other indicators in the health statistics data as regressors to predict specific indicators one by one. By including these variables, the model can capture their impact on the target variable and improve the accuracy of predictions.

$$y(t) = g(t) + s(t) + h(t) + f_1(t) + f_2(t) + ... + f_n(t) + \epsilon(t) \qquad (3)$$

By incorporating relevant external factors, the multivariate model can capture additional information and dependencies that impact the target variable. This can lead to more accurate and reliable predictions. Including additional variables provides insights into the

factors driving the target variable's behavior. It enables a better understanding of the system's relationships and dependencies among different variables. This also allows for customization based on the specific requirements of the forecasting problem. But to incorporate multivariate forecasting, we also found additional complexity, such as complex data preprocessing, feature selection, and potential correlation considerations.

The code to replicate this study can be found at:
https://github.com/iupui-soic/WB-cluster-forecast.

## 4  RESULTS

### 4.1  Clustering

With a distance threshold set at 5, our cluster dendrogram (Fig. 1) presented nine (9) visually distinct clusters.

The Silhouette score, a measure used to evaluate the clusters and the countries within the nine clusters, is displayed in Table 2.

**Table 2: CLUSTERED COUNTRIES AND EVALUATION METRIC**

| Cluster # | Cluster Silhouette Score | Countries |
|---|---|---|
| 1 (European Countries) | 0.2914 | Bulgaria, Belarus, Czechia, Estonia, Croatia, Hungary, Lithuania, Latvia, Slovenia, Ukraine |
| 2 (European, North American, Oceanian Countries and Japan) | 0.4851 | Australia, Austria, Belgium, Canada, Switzerland, Germany, Denmark, Spain, Finland, France, United Kingdom, Greece, Ireland, Iceland, Italy, Japan, Luxembourg, Netherlands, Norway, New Zealand, Portugal, Sweden, United States |
| 3 (East & West African, South Asian and Other Countries) | 0.4227 | Benin, Bangladesh, Congo, Comoros, Eritrea, Ghana, Gambia, Haiti, Cambodia, Madagascar, Mauritania, Nepal, Pakistan, Senegal, Togo, Yemen |
| 4 (Southern African Countries) | 0.2484 | Botswana, Lesotho, Namibia, Eswatini |
| 5 (African Countries) | 0.3309 | Burundi, Burkina Faso, Cameroon, Ethiopia, Kenya, Liberia, Mali, Mozambique, Malawi, Niger, Nigeria, Rwanda, Sierra Leone, Chad, Tanzania, Uganda, Zambia |
| 6 (Ensemble of Countries from Different Regions) | 0.6693 | Albania, Argentina, Armenia, Bahamas, Bosnia and Herzegovina, Brazil, Barbados, Chile, Colombia, Costa Rica, Cuba, Cyprus, Georgia, Israel, Jamaica, Kazakhstan, Sri Lanka, Moldova, Malta, Mauritius, Panama, Singapore, Seychelles, Thailand, Uruguay |
| 7 (Large Economy Countries in Asia) | 0.5667 | China, India |
| 8 (Middle Eastern Countries) | 0.6597 | Bahrain, Kuwait, Oman, Qatar, Saudi Arabia |
| 9 (Ensemble of Countries from Different Regions) | 0.3282 | Azerbaijan, Belize, Bolivia, Algeria, Ecuador, Egypt, Fiji, Guatemala, Guyana, Indonesia, Iran, Jordan, Kyrgyz Republic, Kiribati, Lebanon, Morocco, Maldives, Mexico, Myanmar, Mongolia, Malaysia, Peru, Philippines, Paraguay, Solomon Islands, El Salvador, Turkmenistan, Tonga, Tunisia, Uzbekistan, Vietnam, Vanuatu |

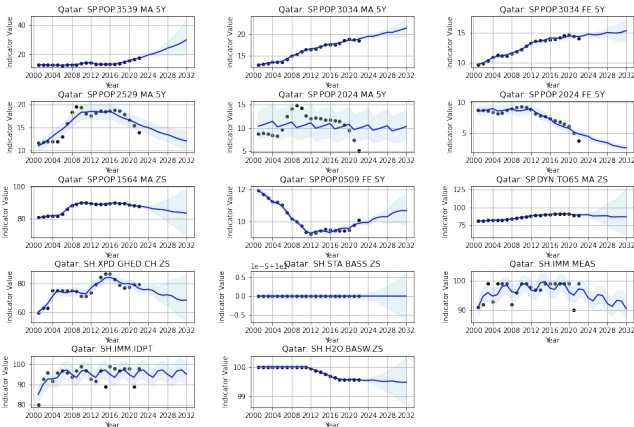

**Figure 2:** Time-series Yearly Data and Future Forecasts for Qatar using Univariate Prophet Model

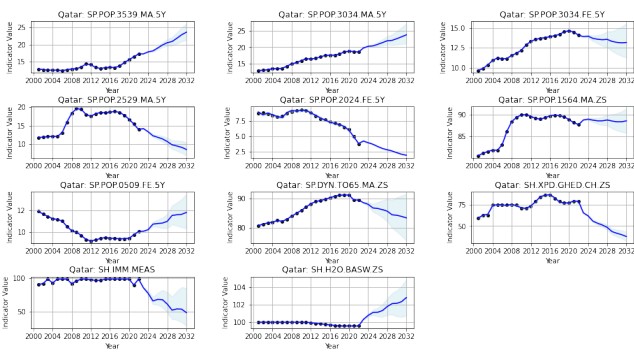

**Figure 3:** Time-series Yearly Data and Future Forecasts for Qatar using Multivariate Prophet Model

### 4.2  Feature Relevance

We analyzed correlations between the features. If an indicator demonstrated a strong positive or negative correlation with any other indicators in the dataset, we excluded it. We retained only those indicators that didn't correlate highly with others. This process yielded 15 indicators out of the original 26 in Cluster-8 shown in Table 1.

### 4.3  Time-Series Forecasting

Our secondary objective was to apply a multivariate time series forecasting Prophet model to the significant indicators of the five countries within a cluster [35]. A preliminary statistical test highlighted similarities in the indicators' values for the year 2000.

*4.3.1  Outcome of Feature Reduction.* Due to many missing values, we excluded two features identified through feature importance. We also removed nine indicators that exhibited a high correlation with other significant features and one indicator that displayed negative values, which was unsuitable for logarithmic transformation. Consequently, we proceeded with univariate forecasting for the remaining 14 indicators.

**Table 3: ACCURACY METRICS FOR THE FORECASTED INDICATOR VALUES AMONG THE COUNTRIES**

| Indicators | RMSE | | MAPE | | $R^2$ | | Adjusted $R^2$ | |
|---|---|---|---|---|---|---|---|---|
| | Prophet (Avg±SD) | LSTM (Avg±SD) | Prophet (Avg±SD) | LSTM (Avg±SD) | Prophet (Avg±SD) | LSTM (Avg±SD) | Prophet (Avg±SD) | LSTM (Avg±SD) |
| Population ages 30-34, male | 0.0001 ± 0.0001 | 0.5941 ± 0.3203 | 0 ± 0 | 0.0401 ± 0.0259 | 1 ± 0 | 0.5997 ± 0.3132 | 1 ± 0 | 0.5497 ± 0.3523 |
| Population ages 30-34, female | 0.0001 ± 0.0001 | 0.2563 ± 0.0961 | 0 ± 0 | 0.0216 ± 0.0109 | 1 ± 0 | 0.6592 ± 0.468 | 1 ± 0 | 0.6166 ± 0.5265 |
| Population ages 35-39, male | 0.0002 ± 0.0002 | 0.3445 ± 0.1631 | 0 ± 0 | 0.0259 ± 0.0093 | 1 ± 0 | 0.6581 ± 0.2856 | 1 ± 0 | 0.6154 ± 0.3213 |
| Population ages 25-29, male | 0.0059 ± 0.0127 | 1.1566 ± 0.7031 | 0.0006 ± 0.0013 | 0.071 ± 0.0374 | 1 ± 0 | 0.6031 ± 0.252 | 1 ± 0.0001 | 0.5535 ± 0.2835 |
| Population ages 20-24, female | 0.0287 ± 0.0637 | 0.4546 ± 0.3414 | 0.0032 ± 0.0072 | 0.0421 ± 0.0381 | 0.9979 ± 0.0046 | 0.5067 ± 0.3441 | 0.9956 ± 0.0097 | 0.445 ± 0.3871 |
| Population ages 15-64, male | 0.001 ± 0.0012 | 1.2822 ± 0.8086 | 0 ± 0 | 0.0143 ± 0.0092 | 1 ± 0 | 0.4109 ± 0.6021 | 1 ± 0 | 0.3372 ± 0.6774 |
| Population ages 05-09, female | 0.0001 ± 0.0001 | 0.5177 ± 0.1904 | 0 ± 0 | 0.0458 ± 0.0212 | 1 ± 0 | 0.0855 ± 1.1177 | 1 ± 0 | -0.0288 ± 1.2574 |
| Survival to age 65, male | 0.001 ± 0.0007 | 1.1749 ± 0.7324 | 0 ± 0 | 0.0125 ± 0.0089 | 1 ± 0 | 0.5497 ± 0.6035 | 1 ± 0 | 0.4935 ± 0.6789 |
| Domestic general government health expenditure | 0.4999 ± 0.498 | 2.3871 ± 1.0832 | 0.0058 ± 0.0059 | 0.0282 ± 0.015 | 0.9681 ± 0.0409 | 0.4775 ± 0.2928 | 0.933 ± 0.0859 | 0.4122 ± 0.3294 |
| Immunization, measles | 0.0009 ± 0.0008 | 1.0328 ± 0.6968 | 0 ± 0 | 0.0086 ± 0.0054 | 1 ± 0 | 0.2123 ± 0.2276 | 1 ± 0 | 0.1139 ± 0.256 |
| People using at least basic drinking water services | 0.0008 ± 0.0006 | 0.2138 ± 0.3582 | 0 ± 0 | 0.002 ± 0.0035 | 0.7997 ± 0.4471 | 0.5849 ± 0.3723 | 0.5794 ± 0.9388 | 0.533 ± 0.4189 |

*4.3.2 Statistical Testing on the Existing Indicator Values.* We performed the Kruskal Wallis test on the values of the 15 indicators for the countries within the clusters. The resulting p-values were all greater than 0.05, suggesting no statistically significant differences among the values of the indicators within the clustered countries. Since these indicators demonstrated similar values across countries, we continued with time series forecasting.

*4.3.3 Univariate & Multivariate Prophet.*

*Future Dataframe.* Univariate Prophet modeling produced reliable predictions for most indicators, yielding low RMSE & MAPE and better $R^2$ value. However, three indicators demonstrated inferior $R^2$ values compared to others, leading us to exclude them from the multivariate models. These indicators were: Population ages 20-24, male (% of male population), Immunization, DPT (% of children ages 12-23 months), and People using at least basic sanitation services (% of the population).

*Future Forecasts.* The multivariate Prophet model generated forecasts for each of the 11 indicators under consideration. In each forecast, the multivariate model included 10 additional regressors corresponding to the other 10 indicators, serving as predictors excluding the target indicator. The accuracy metrics for the multivariate models are detailed in Table 3. The univariate forecasting model predicted 15 indicators for a sample country (Qatar), and the multivariate model predicted 11 indicators (see Fig. 2 and Fig. 3 respectively). These figures illustrate the multivariate Prophet model's superior forecasting performance. The combined forecasts for the clustered countries (Bahrain, Kuwait, Oman, Qatar, and Saudi Arabia from the year 2000 to 2031 for all 11 indicators are illustrated in Fig.4 with continuous error bar plots. The differences in the indicators in the future years can be seen in Fig. 4

*4.3.4 Statistical Analysis on the Forecasting.* The future forecasted indicator values also showed statistically significant differences (p<0.05) among the countries, highlighting that the forecasted trajectory of the countries might be changing in the future based on the already changing nature of predictors. Using univariate forecasting, such modeling would not have been possible.

## 5 DISCUSSION

Health econometrics analyses have traditionally relied on cross-country surveys like the National Family Health Survey (NFHS) and the Demographic Health Survey (DHS). They often employ logistic regression and other statistical techniques for comparing countries [20, 33]. Among unsupervised statistical approaches, I-distance [11, 12] has been utilized for ranking purposes, including countries based on health indicators. However, our study presents the potential of enhanced clustering machine learning techniques for managing multiple related variables, particularly for large datasets. [21].

Notably, certain clusters, such as Cluster-4 and Cluster-8, display geographical and cultural similarities. The cluster linkage cutoff would need to be significantly lowered to establish more readily apparent similarities within each cluster. However, this could lead to fewer predictor indicators, affecting our features of importance. If we expand the indicators used in feature selection, we risk complicating the model and reducing its interpretability. [16].

Other clustering algorithms, especially spectral clustering, while a powerful technique in certain cases, may not always be the most appropriate choice. It operates based on graph theory principles and requires constructing a similarity matrix and computing eigenvectors, which can be computationally expensive and memory-intensive for larger datasets. Spectral clustering also contains a stochastic factor which was avoided by using hierarchical clustering

Given the size and nature of our dataset, hierarchical clustering with Ward's method proved to be a more scalable and efficient option. It aligns well with our goals of exploring hierarchical patterns and capturing diverse cluster shapes in the health and nutrition dataset. Hierarchical clustering also provided meaningful insights into the health indicators across countries. Along with this, Logarithmic scaling on the dataset provided less mean squared error on a whole in the prediction of the future features' values compared to Min-Max scaling.

While our models present robust and meaningful findings, they also highlight some challenges that need to be considered in future studies. A critical point is the trade-off between the granularity of clustering and the complexity of multivariate models. While deeper clustering might yield more nuanced insights, it can also reduce

the number of predictor indicators and increase model complexity. It calls for a balanced approach to ensure the interpretability and practical utility of the models.

Additionally, our multivariate forecasting model is predicated on current and past trends. The dynamic nature of health indicators and their susceptibility to various external factors such as political changes, economic fluctuations, or global health crises, might alter these trends significantly. Future research must consider these potential disruptions and explore methods to account for such unpredictability.

Further, we could determine certain associations by understanding the identification of statistical differences amongst features that we obtained after analysis and predictions from a multivariate model. Viewing Fig.4i, where Qatar's future prediction on health expenditure seems to decline, and Fig.4j also indicates a decline in immunization. Similar declines are seen in female population ages who are potentially at a maternal period (Fig.4c and 4e). We drew validating conclusions that our multivariate prophet model determines the reliance of a feature on another feature for a country [22]. This can aid the several health assessment research associated with various indicators such as work by Amoatey et al. [3].

Recognizing these trends and connections could guide policymakers or health practitioners toward effective strategies for improving overall health outcomes. Moreover, our predictions consider various population age groups, offering a comprehensive perspective on health prospects [9]. Our study's application of multivariate forecasting allowed us to predict future health outcomes based on current trends and patterns. This model has allowed us to project possible trajectories for various health indicators in the Middle Eastern countries cluster, aiding in long-term strategic health planning for the region. The associations identified between different features underline the interconnectedness of health outcomes, signaling the necessity for an integrated approach to healthcare policy.

## 5.1 Limitations

This study has its limitations. Although we selected 26 indicators from the World Bank dataset's total of 128, not all could be incorporated into our multivariate prediction model. For example, the Population Growth indicator was excluded because it contained negative values incompatible with logarithmic transformation. However, our

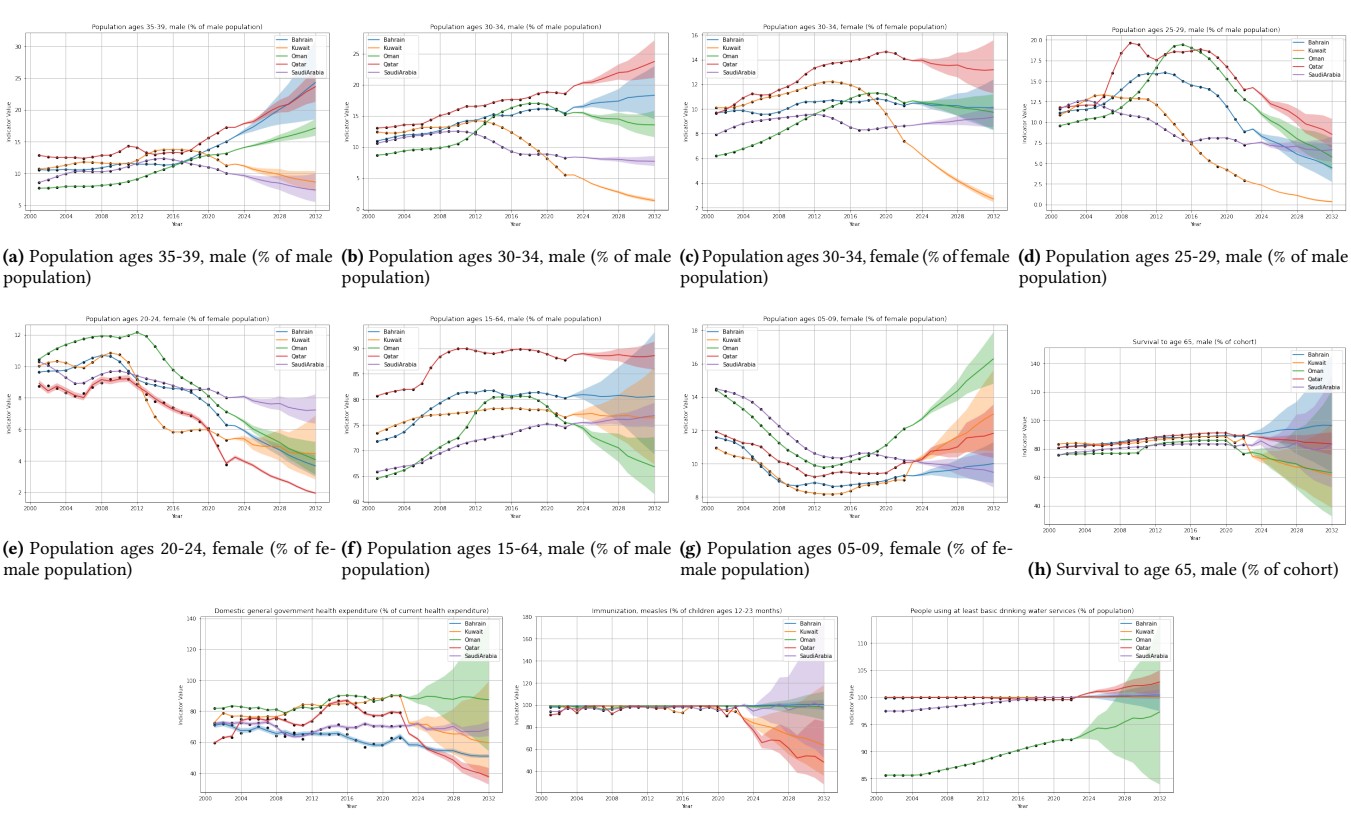

**(a)** Population ages 35-39, male (% of male population)

**(b)** Population ages 30-34, male (% of male population)

**(c)** Population ages 30-34, female (% of female population)

**(d)** Population ages 25-29, male (% of male population)

**(e)** Population ages 20-24, female (% of female population)

**(f)** Population ages 15-64, male (% of male population)

**(g)** Population ages 05-09, female (% of female population)

**(h)** Survival to age 65, male (% of cohort)

**(i)** Domestic general government health expenditure (% of current health expenditure)

**(j)** Immunization, measles (% of children ages 12-23 months)

**(k)** People using at least basic drinking water services (% of population)

**Figure 4: Forecasts of each indicator for five clustered countries**[*]
[*]Blue forecast lines are for Bahrain; Orange forecast lines are for Kuwait, Green forecast lines are for Oman, Red forecast lines are for Qatar, and Purple forecast lines are for Saudi Arabia

model's predictions could be significantly influenced by the inclusion of this indicator. Similarly, other omitted indicators could have offered additional insights into overall health outcomes.

## 5.2 Future Work

Future work could involve constructing a more informative model with an expanded set of features or a larger cluster of countries. Techniques like Neural Prophet [37], DeepAR [30], or even simpler models like Random Forest Regressor [10] could be explored. Alternative approaches to constructing future data frames, such as Auto ARIMA, could yield more reliable results.

## 6 CONCLUSION

In conclusion, our study has identified key factors influencing health outcomes in selected Gulf Cooperation Council (GCC) countries (Bahrain, Kuwait, Oman, Qatar, and Saudi Arabia). We highlighted the importance of population wellness and age-specific strategies in healthcare management and disease prevention. Our method involved data preprocessing, clustering using Ward's method, feature selection, and time series forecasting with multivariate Prophet. This research provides a comprehensive approach to health data analysis, identifying crucial health outcome influencers, and delivering actionable insights for policymakers and healthcare professionals using machine learning and forecasting techniques.

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
