# OpenReview forum: "Hierarchical Clustering and Multivariate Forecasting for Health Econometrics"
_KDD.org/2023/Workshop/epiDAMIK — KDD 2023 Workshop epiDAMIK_

### Official Review · Reviewer_r7Ax · 2023-06-16
**Reject**

**Rating:** 1
**Confidence:** 4

**Review:**

In this paper, the authors used hierarchical clustering to group 131 countries into several clusters and then performed a time-series forecasting for the cluster consisting of several Middle Eastern countries. While forecasting socio-economic and health indicators is important for policymaking, the methods used in this study are relatively simple. I have a few concerns about the study and results.
1. Both the clustering and forecasting methods are off-the-shelf approaches. It is not clear the methodological novelty of this study. For instance, time-series forecasting is widely used in other studies.
2. There was no comparison of the forecasting method with other approaches. There should be more accurate forecasting methods, and the authors did not establish the advantage of the current method in this study.
3. Lack of details.  Many technical details were not provided in the manuscript. For instance, what features were included in the dataset? What additional variables were used in the multivariate forecasts, and how to select those variables? How did the authors select the prediction target variables in Table 3? Was it because the forecasting method worked better for those targets?
4. In Eq. (3), the prediction of y(t) needs the input of exogenous variables in the future, which is not available when the forecast is generated. How to solve this issue? How to decide which exogenous variables to include?

---

### Official Review · Reviewer_x9gB · 2023-06-28
**Interesting analysis**

**Rating:** 4
**Confidence:** 3

**Review:**

## Clarity

This paper and proposed method is easy to follow.

## Quality

The analysis is well-motivated and fully delivered the idea.

## Originality

This is original work with interesting problem.

## Significance

The work is significant.

## Pros:

- Well-written, and clearly delivers the ideas, proposed method, and results.

- The analysis is interesting to me.

- The authors are well aware of the limitations of the proposed method.


## Cons:

- The way authors get feature importance is not clear.

- Authors may consider using different methods for multivariate time-series forecasting such as MLP, LSTM, …

- Authors did not include similar analysis for the univariate case to highlight the benefit of the multivariate model, although authors remove some features based on the performance of univariate models.

- The variance for future forecasting results are high, then the conclusion is a bit uncertain (besides the mentioned factors like political changes, economic fluctuations, …)

---

### Official Review · Reviewer_e92c · 2023-06-29
**Hierarchical Clustering and Multivariate Forecasting for Health Econometrics- Review**

**Rating:** 4
**Confidence:** 4

**Review:**

**Summary:**

This study uses clustering and time series forecasting to create retrospective scenarios for forecasting the long-term impact of changes in socio-economic indicators on health indicators. Firstly, the authors used hierarchal clustering to group countries based on socio-economic indicators based on the World Bank Health Statistics and Nutrition dataset. Following that, the authors performed time series analysis to predict the values of the different indicators using  the multivariate prophet model on the countries which appear in one of the groups. This led to valuable insights about the dynamics in the future.

**Strong Points:**

- The clusters constructed are interesting. Cluster 1 seem to be a whole group of Eastern European Countries which are geographical neighbors. Cluster 3 countries are not geographically close but are developing nations. Cluster 4 & 5 seem to be African countries and so on...
- The authors perform a thorough literature review which provides a good platform to evaluate the significance of this study.
- This is a well written paper. The explanations provided are good, the figures are well made and it surely applies a variety of methods. This can surely add to the technical contributions of this work.

**Weak Points:**

- Why hierarchal clustering? Spectral clustering is also a good method, right? The authors need to mention their motivation behind using hierarchal clustering.
- Retrospective Interpretations of clusters are needed as the relation in some of them are not that obvious. For example, what is the relation between the countries that appear in cluster 2? It's not that clear.
- In Figure 4, some of the indicator forecasts have a high level of uncertainty more than the others. Exploring what is causing this is extremely valuable but is sadly missing.

**Suggestions:**

- I understand that logarithmic scaling gave better performance. However, one of the lim imitations mentioned of forecasting the Population Growth indicator could be easily done by using Min-Max scaling. By any means, the performance could have been reported.
- What is the significance of a threshold of 0.815 in section 3.4? Was it used in prior works?